# Application of a DC Distribution System in Korea: A Case Study of the LVDC Project

**Juyong Kim \*, Hyunmin Kim, Youngpyo Cho, Hongjoo Kim and Jintae Cho**

Smart Power Distribution Lab., KEPCO Research Institute, 105 Munji-Ro, Yuseong-Gu, Daejeon 34056, Korea;
hm.kim@kepco.co.kr (H.K.); yp.zo@kepco.ko.kr (Y.C.); hongjoo.kim@kepco.co.kr (H.K.);
jintae.cho@kepco.co.kr (J.C.)

\* Correspondence: juyong.kim@kepco.co.kr; Tel.: +82-42-865-5951

**Abstract:** With the rapid expansion of renewable energy and digital devices, there is a need for direct current (DC) distribution technology that can increase energy efficiency. As a result, DC distribution research is actively underway to cope with the sudden digitization and decentralization of load environment and power supply. To verify the possibility of DC distribution, Korea Electric Power Corporation (KEPCO) Research Institute made a DC distribution system connected with a real power system in Gwangju. The construction of the demonstration area mainly includes design of protection and grounding systems, operating procedures of insulation monitoring device (IMD), and construction of power converters. Furthermore, this paper goes beyond the simulation and the lab testing to apply DC distribution to a real system operation in advance. It is designed as a long-distance low-loaded customer for rural areas and operated by the DC distribution. In addition, safety and reliability are confirmed through field tests of DC distribution elements such as power conversion devices, protection and grounding systems. In particular, to improve the reliability of non-grounding system, the insulation monitoring device was installed and the algorithms of its operational procedures are proposed. Finally, this paper analyzes the problems caused by operating the actual DC distribution and suggests solutions accordingly.

**Keywords:** DC distribution system; AC/DC converter; protection; grounding system; insulation monitoring device (IMD)

---

## 1. Introduction

The share of DC power consumption in PCs, TVs, DC buildings, Internet Data Centers (IDCs), and DC homes is expected to increase. In particular, EPRI in the United States estimates that digital devices will account for 50% of the world's total DC load in 2020. In addition, due to the expansion of renewable energy, such as photovoltaics (PV) generation and fuel cells, there is a need for a new high-quality electricity service, such as DC distribution service technology. In existing AC distribution systems, the power conversion to DC was required to take advantage of the distributed resources or the DC consumers, which resulted in additional cost and power losses. However, in the case of DC distribution systems, the power conversion step can be omitted to enable a more efficient and economical operation, compared to the existing AC distribution system, when the DC distributed resources and DC consumers are connected [1]. Previous studies have confirmed the economics of the Direct Current (DC) distribution system [2]. In addition, the DC distribution system can effectively control voltage using the power conversion device instead of the transformer, and it can limit the fault current in case of an accident at the load side and prevent accidents from spreading to the main grid. However, there are still difficulties for DC distribution system operation and there is insufficient research on connecting with the actual grid system. The accurate detection and location of fault results

in fast restoration of the system are needed [3]. In previous studies, the lab test was performed prior to applying the DC distribution to the actual grid system, confirming the tremendous potential of the DC distribution system [4]. In this paper, the DC distribution was applied to the actual grid system and confirmed the possibility of its commercialization. The construction of a demonstration area has been verified by simulations and field tests. Also IMD, the DC distribution fault detecting system, has been installed and its operating algorithms are proposed.

## 2. Design of DC Distribution System

This paper represents the design of the DC distribution system for the long-distance low load in rural area. Because the DC distribution system is entirely different from the existing AC distribution system, it should begin with the fundamental reviews of the components to supply DC power to the customers [5–7]. This chapter represents the design of DC distribution systems, including protection and grounding.

### 2.1. Grounding and Protection

The grounding and protection of the DC distribution system in constructing the DC distribution system are an indispensable factor for the stable operation of the system, together with the safety of the users and the protection of the equipment. This chapter describes grounding and protection methods to be applied to the DC distribution system.

### 2.1.1. Grounding

The classification of grounding, which saves a human life and protects properties from electric shock that could cause fire and lightning, follows the protection method of IEC 60364, IT(Isolate-Terra), TN(Terra-Neutral), and TT(Terra-Terra) methods. Each grounding method is classified in accordance with the purpose and the use. In this study, the IT grounding system, which is deemed to be the most suitable for the DC distribution system, is applied to the plan. In the case of the IT grounding method, none of the power lines are grounded to earth, and only the consumer side conductive enclosure is grounded. Unlike other grounding methods, the IT grounding method is suitable for the DC distribution system because it does not cause any electrolytic corrosion appearing on the earth pole [8].

In order to check the safety of the IT grounding system, the ground fault simulation, which is DC distribution with a 750 Vdc of 1 km line and IT grounding system, was performed (Figure 1). When a ground fault is simulated on the pole, no-fault current flows and the rise of ground potential is zero. In practice, however, a slight voltage rise may occur due to the isolation between the system and the ground. In the case of IT grounding, the ground potential rise does not occur when a ground fault occurs. However, when the second ground fault occurs, a fault current flows to the ground, and the rise of ground potential is very steep. According to reference [9], the rise of ground potential is less than 10 mV when the insulation resistance is 1 MΩ, and it is considered as a ground fault when the insulation resistance value is less than 20 Ω. Figure 2 shows the relation between the pole voltage, insulation resistance, and touch voltage of the DC distribution system of the IT ground. As can be seen from the figure, the IT grounding system hardly causes the ground potential rise, even if the insulation level is low. The ground fault in the underground and overhead lines are then simulated. The graph shown in Figure 3a shows the simulation result of the contact voltage at the ground fault in the underground line. In the event of a fault, a voltage close to the terminal voltage is generated and is decreased in 4 μs, falling below the maximum contact voltage of 120 Vdc. After 10 μs, the contact voltage drops below 1 Vdc. Since the period of a dangerous level of voltage generation is very short, the danger to the human body is very low.

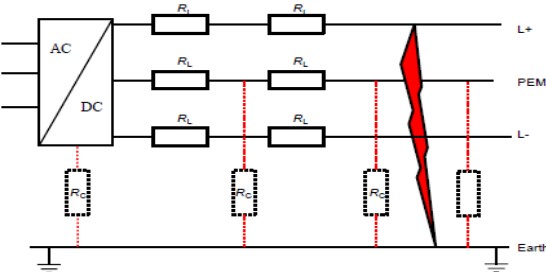

**Figure 1.** Ground Fault IT(Isolate-Terra) System.

Figure 3b shows the ground fault in the overhead line. In the case of an overhead line, the resonance of the contact voltage occurs at the beginning of the fault due to the inductance of the line. Theoretically, the maximum impulse voltage should be the same as the pole voltage at the beginning of the fault. However, the simulation results show that the initial impulse voltage is relatively low because of the low capacitance of the line. However, in the case of the overhead line, in contrast to the underground line, the resonance phenomenon of contact voltage attributed to fault occurred for a relatively long 100 μs. However, the voltage exceeding 120 Vdc dissipates almost instantaneously with the occurrence of the fault. Therefore, one can conclude that safety is not a concern in this case. Based on international standards, it is acceptable for the human body to be exposed to 800 Vac for 0.04 s. The simulation results of the aforementioned overhead line and the underground line all disappeared within 100 μs. Therefore, it is considered that the discharge current due to the total capacitance between the DC line and the ground in the IT system moves quickly, and no danger to the human body is anticipated. Given that the TT and TN grounding methods connect one power line to the ground through a protective conductor, a ground fault in any one of the power lines causes a fault current to be generated because of the closed circuit between the power sources by the protective conductor. The IT grounding method, on the other hand, is advantageous even in the case that the one-wire ground fault in the power line does not cause an interruption of the power or the load supply because it does not constitute a closed circuit. However, even when the IT grounding method is used, if a two-wire ground fault occurs, a closed circuit is formed and causes a ground fault. Therefore, in order to secure the advantages of the IT grounding method and prevent electric shock accidents, it is important to detect the accident on the one-wire ground fault and take measures to prevent a two-wire ground fault. The IT grounding method is characterized by a decrease in the insulation resistance between the power line and the ground in the event of a ground fault. Such deterioration of insulation resistance leads to insulation breakdown, which can cause damage due to electric shock and fire. To prevent such failure, the IEC 61557 specifies that an insulation monitoring device (IMD) should be installed in an IT grounding system environment to protect facilities and the human body from one-line ground fault [10]. Therefore, an IMD is installed and applied in this study.

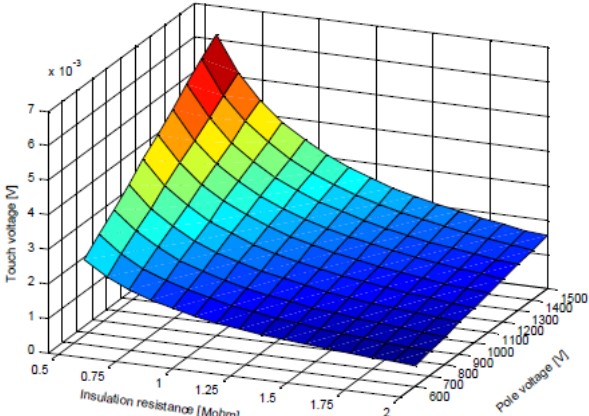

**Figure 2.** Correlation diagram of pole voltage, insulation level, and contact voltage in IT ground.

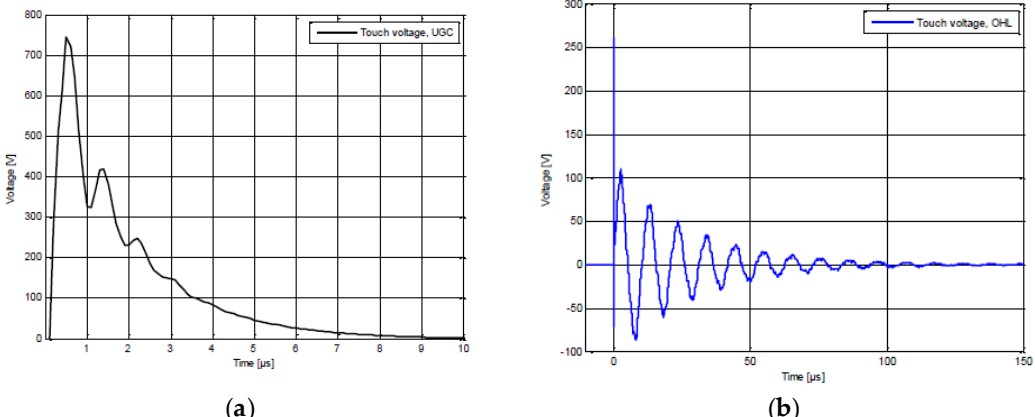

**Figure 3.** Simulation of ground fault in DC distribution line of IT ground: (**a**) underground line; (**b**) overhead line.

### 2.1.2. Fault Detection Device

In the case of non-grounded systems, it is difficult to detect grounding faults by a conventional method. Therefore, real-time analysis of the insulation resistance in the energized-line is necessary. Thus, an IMD is installed.

IMD is commonly used to measure insulation resistance in closed circuit rail or medical devices [11]. Currently, the development of IMD technology for the distribution line is in the early stage, and there is no application for outdoor use. Furthermore, it is necessary to study operation standards and procedures. Therefore, this paper confirms the applicability of IMD and presents the operational procedures. It is necessary to set appropriate alarm value to detect ground faults while operating the distribution system and coping with system stoppage or re-input according to the situation. However, there is currently no standard for IMD alarm set-points for DC distribution lines. However, in "Selection and erection of electrical equipment—isolation, switching and control" of IEC 60364-5-53 Annex H, the pre-warning is specified as 100 Ω per 1 V and the warning as 50 Ω per 1 V. The setting value of IMD is recommended to be set to 1.5 times this value. It is essential to select an appropriate alarm set-point because the IMD for the DC distribution line monitors the insulation of the DC distribution line and the associated power converter. Therefore, a flow chart of the operation procedure was proposed for determining the IMD alarm setting value required in the field application (refer to Figure 4). In the case of alarm 1 of IMD, be cautious of the possibility of decreased insulation resistance in the line, while the alarm 2 should immediately trigger stopping of the operation, followed by an inspection. This is to prevent possible grounding caused by animals or humans, which can lead to secondary failure, which may lead to deadly accidents. In this study, the alarm value is set by measuring the real-time insulation resistance value according to the proposed operating procedure for the DC distribution to be applied to the actual grid system.

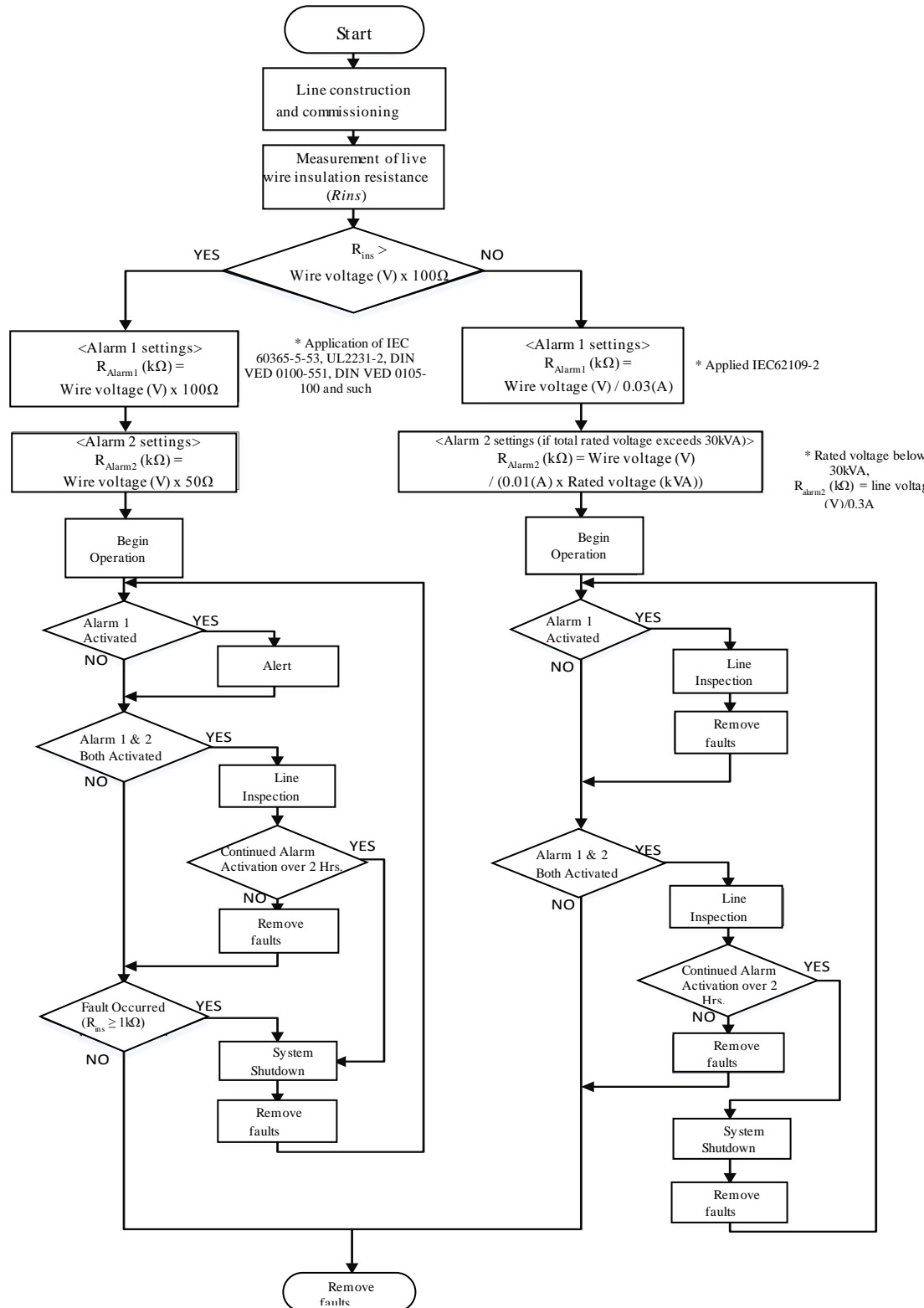

**Figure 4.** Proposed insulation monitoring device (IMD) operating procedures.

### 2.1.3. Protection

The purpose of the power system protection is to secure people and equipment by eliminating faults or limiting voltage and current below a certain level. If a failure occurs in the DC distribution

system, the protection operation should be performed for the faulty part, and the fault should be detected, even when it does not require any protective operation [12,13].

In the DC distribution system, the protection system can be divided into three parts: power supply rectifier, DC distribution line, and consumer. The protection system should be determined by the system grounding method, voltage level, network configuration, and power converter structure. Protection of the system shall meet the following requirements.

- The risk level should not be higher than the existing AC system.
- Faults should be detected and removed.
- Fault section should be separated.
- There should be no breakdown of equipment due to line failure.
- Safety should not be threatened by equipment failure.

The protection requirements of the above system can be satisfied by protective devices such as converter protection system, circuit breaker (CB), overcurrent/overvoltage relay, surge protection device (SPD), IMD, grounding system, electrical isolation, and insulation cooperation.

As shown in the Figure 5, the protection system of the DC distribution system was designed. First of all, a surge protection device needs to be installed on the primary side of the transformer to protect against overvoltage which can flow through the high-voltage line. The transformer does not need to be grounded on the secondary side because the flow of current through the earth can cause permanent failure. A DC circuit breaker (DCCB) was installed on the rectifier output side to prepare for overcurrent and short circuit faults that may occur in each pole. In addition, the DCCB can be used when the rectifier between the high voltage line and the DC line needs to be independent, allowing for the independent operation of the lines on each end. At this moment, the converter system can supply limited fault current when the fault occurs and control the DCCB by the measured fault results. Since the use of the IT grounding system, the IMD is installed to monitor grounding faults. If a grounding fault occurs in the DC line, the insulation monitoring device controls the DCCB, and if the grounding situation is not resolved, the DCCB between the transformer and the converter system is controlled to cut off the power supply. In addition, an SPD is installed to protect the converter system and the line against the DC line overvoltage input. Likewise, an SPD is installed in front of the customer's side to provide protection against overvoltage inflow to the customer's network and the inverter. In addition, a DCCB is installed in front of the customer's inverter to protect the DC line from the fault of the customer's network and the inverter. The inverter limits the amount of current passing through at the time of the fault to ensure the normal operation of the CB, while the CB installed on the customer's power supply line performs the protection operation if the operating characteristics curve in the event of failure.

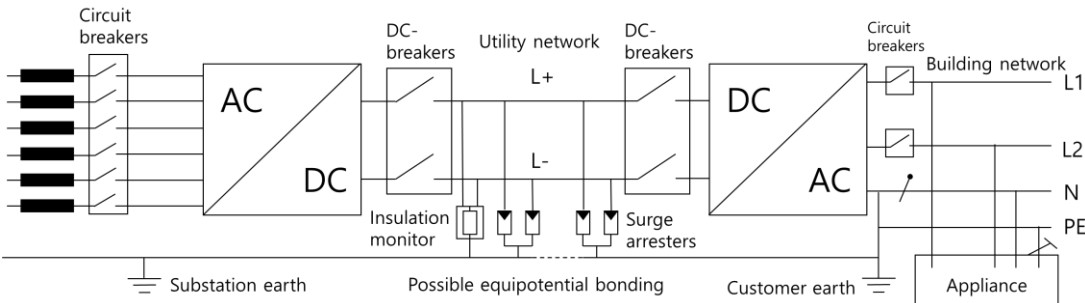

**Figure 5.** Design of protection system for DC distribution.

## 3. Field Test

Before applying the DC distribution element to the field, the application test of the IMD and the converter system device is carried out on the DC distribution line installed in the power test center.

The power test center is a demonstration site similar to the environment in which actual grid system are installed. The center was constructed to improve the reliability of a device under test (DUT) and to confirm its performance (Figure 6 and Table 1). A 70SQ CV cable on the demonstration site, a long-range 700 m power line, was used in the test.

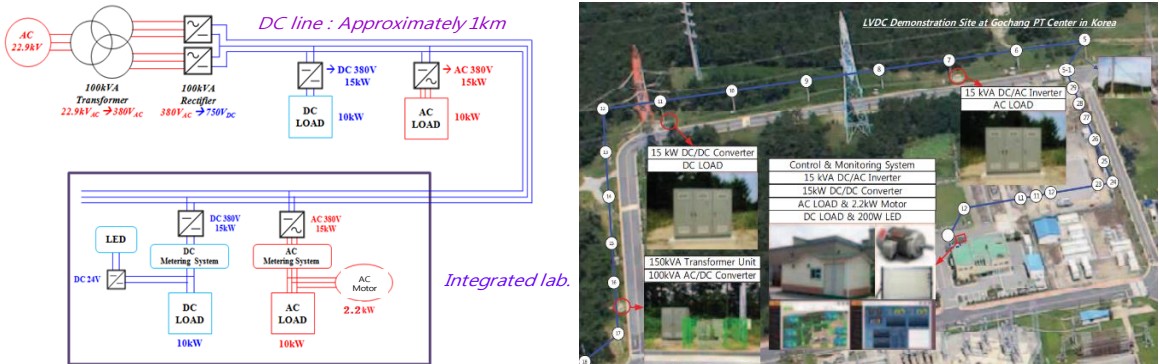

**Figure 6.** Configuration of DC demonstration site.

**Table 1.** Specifications of component in power test center.

| Item | Capacity | Topology |
|---|---|---|
| AC/DC Converter (Rectifier) | 100 kW | 2-level type |
| DC/DC Converter (2EA) | 15 kW | Buck-Boost type |
| DC/AC Converter (2EA) | 15 kW | 2-level type |
| Transformer | 100 kVA | - |

### 3.1. Fault Detection

To verify the IMD, at the DC demonstration site, the application test was divided into two cases.

(a)   Case 1

The condition of first IMD application test is shown in Figures 7 and 8. As shown in the figures, the P-pole grounding fault occurred in the middle of the power line, and the measurement position of the IMD is the output side of the AC/DC converter. Also, the short circuit resistance of the grounding fault simulator was set to 0.632 Ω to limit the grounding fault current to 600 A maximum, and the fault duration was set to 10 s. The IMD generated Alarm 1 and 2 at approximately 5 s after the fault occurred.

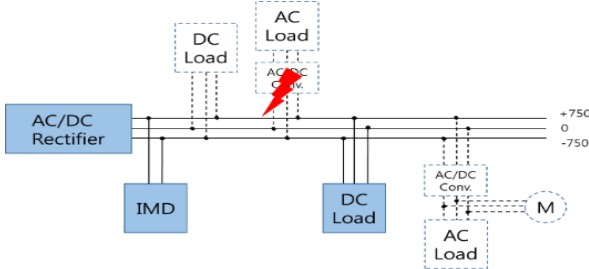

**Figure 7.** Fault detection test case 1.

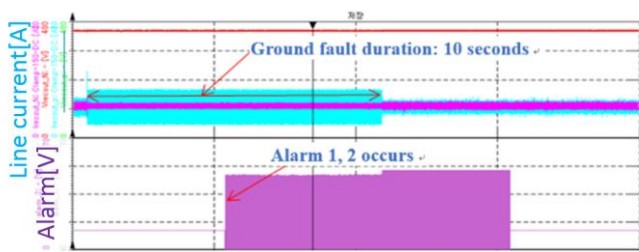

**Figure 8.** Fault detection test case 1 result waveform.

(b)　Case 2

In case 2, the IMD performance test was conducted with varying loads. The configuration is shown in Figure 9. The real-time insulation resistance of the IMD was measured when the IMD was connected to the AC/DC converter output side, and each load of the demonstration site was driven. The insulation resistance of the IMD was measured in real time while the IMD was connected to the output side of the converter to pass the voltage through the line, and all the loads at the demonstration site were applied. Table 2a shows the measured values when the converter is activated in the no-load condition.

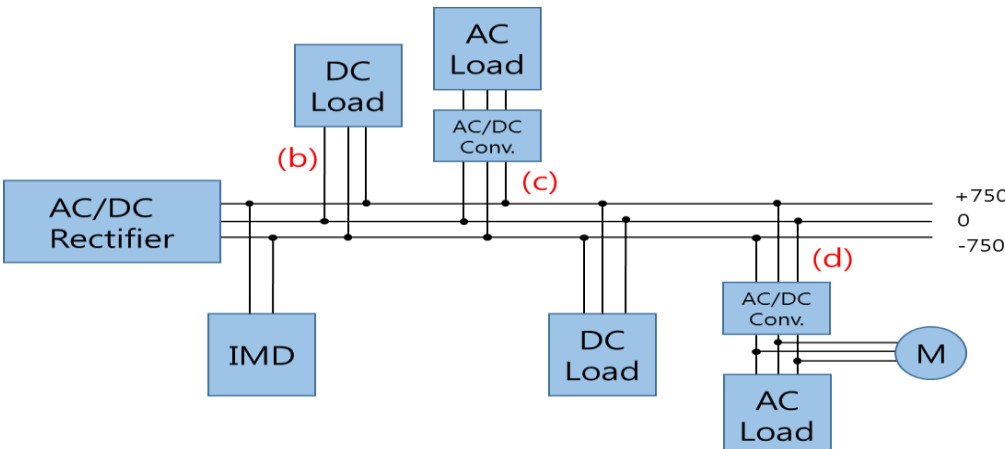

**Figure 9.** Fault detection test case 2.

The real-time insulation resistance value is above the maximum measurable value of 10 MΩ. Case (b)–(d) show the real-time insulation resistance value when the simulated loads corresponding to Figure 9. It shows a healthy level of insulation resistance of the test line and power converters from 225 kΩ to 295 kΩ. Therefore, it was confirmed that the distance and variation of load do not significantly impact the insulation resistance value. Also, it was found that the failure detection time and the failure resolution recognition time were about 5 s, therefore the instantaneous response was impossible.

**Table 2.** Measurement of insulation resistance.

| Case | Insulation Resistance |
|------|----------------------|
| (a)  | >10 MΩ               |
| (b)  | 224 kΩ               |
| (c)  | 253 kΩ               |
| (d)  | 295 kΩ               |

### 4. Construction of Actual DC Distribution Line

In this chapter, applying actual DC distribution studies carried out based on the previous studies are discussed. This chapter established DC distribution system to connect to the actual grid system and confirmed the feasibility of the DC distribution system.

*4.1. Commercial Distribution Line*

The existing long-distance low-load grid line, which supplied high-voltage AC power, is replaced by the DC distribution system. The target line is a long-distance (over 12 spans) and low-load (under 30 kW) aerial bundled cable (ABC), where the possibility of line failure due to contact with trees is high. The location of the target line is 52L1~52L12 at Chungok Reservoir, a Gwangju Jeonnam HQ controlled area.

In, DC distribution, a DC/AC inverter has been used to convert a 13.2 kV high-voltage AC line into DC line. Unlike any conventional transformer, controlling voltage and current at the output stage is possible, as well as the power factor through the internal control algorithm of the converter device. In addition, since the low-voltage line replaces the existing high-voltage line, it is expected that the fault that occurs due to the contact of the trees can be reduced.

*4.2. DC Distribution Equipment*

The significant equipment of the DC distribution system connected to the actual grid in Gwangju are primarily composed of the transformer, converter system, protection system, and monitoring system. The specifications of each device are shown in Table 3. Single-phase AC/DC converter and single-phase DC/AC inverter switching devices use SiC for efficiency improvement [14,15].

**Table 3.** Specifications of DC distribution equipment.

| Item | Pole Transformer | Converter (AC/DC) | Inverter (DC/AC) |
|---|---|---|---|
| Input voltage | 1∅ 13.2 kVac | 1∅ 220 Vac | 750 $V_{dc}$ |
| Capacity | 30 kVA | 30 kW | 30 kW |
| Rated voltage | - | 750 $V_{dc}$ | 220 Vac $\pm$ 5% |
| THD | - | <5% | <5% |
| Efficiency | - | $\geq$97% | $\geq$97% |

The detailed system configuration is shown in Figure 10. In case of a problem with the DC distribution system, the customer can switch to the existing AC system in order to prepare for the long-term power outage. The Cut-Out Switch (COS) shown in Figure 10 and the pole transformer on the load side can supply AC. To supply DC voltage, open the COS (①) for the two lines and put COS (②) to use. On the contrary, to supply AC voltage, put COS (①) to use for the two lines and open the COS (②). The power supply and the load side AC/DC Converter are separated from the transformer to ensure stability. A low-voltage line was constructed below the neutral line in consideration of the induction voltage of the high-voltage line. Also, to build a non-grounding (IT) system, the grounding of the enclosure through the three-class grounding was enabled, and the IMD for monitoring the grounding fault of the line was installed.

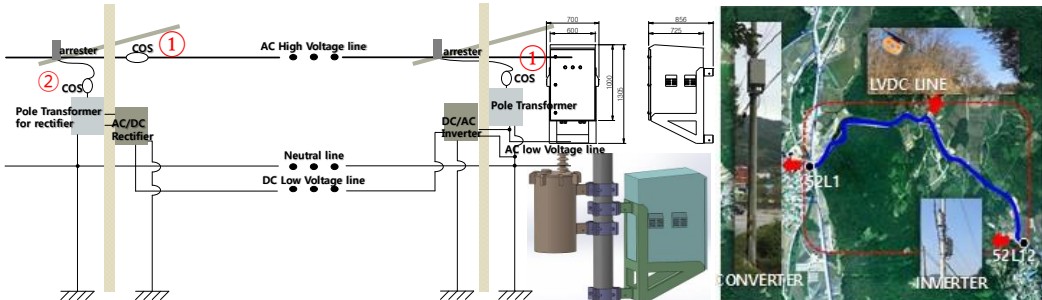

**Figure 10.** Actual DC distribution line configuration.

### 4.3. Operation Results

The DC distribution was applied to the actual gird line to find problems and to improve the performance. Also, by establishing operating data for the pilot lines, design, and operation standards for DC distribution lines were created.

The performance of the AC/DC converter applied to the actual grid line was verified. The average daily data from 14 February 2017 to 30 July 2017 are summarized. Approximately 86,000 data points were collected per day, and the average was calculated. The voltage was very stable, with a fluctuation of 5% over the 6 months. Except for the 23 February 2017 the efficiency remained near 98%.

On 23 February 2017 because the load usage was zero from 06:37:54 to 10:09:14, the daily average efficiency was low. The data showed that the performance of the AC/DC converter system was satisfactory.

As shown in the Section 3.1 "Fault detection," the IMD was installed on the output side of the AC/DC Converter as the IMD installation location does not impact its performance. In particular, this paper tested the feasibility of the outdoor operation of IMD by installing it in an outdoor environment, which is rarely done. The insulation resistance value ($R_{INS}$) of the energized-line was measured upon the installation of the IMD, following the operation manual shown in Figure 4. The measured value was 487 k$\Omega$ so $R_{INS}$ is 750 (line voltage) $\times$ 100 $\Omega$ $\times$ 1.5 = 112.5 k$\Omega$ or more. Therefore, the alarm set value is set as follows.

(i) Alarm 1—pre-warning

R_Alarm1 (k$\Omega$) = line voltage (V) $\times$ 100 ($\Omega$) $\times$ 1.5 = 750 $\times$ 100 $\times$ 1.5 = 112.5 (k$\Omega$)

(ii) Alarm 2—warning

R_Alarm2 (k$\Omega$) = line voltage (V] $\times$ 50 ($\Omega$) $\times$ 1.5 = 750 $\times$ 50 $\times$ 1.5 = 56.25 (k$\Omega$)

Next, Figure 11 is a measure of the IMD resistance value obtained from the actual DC distribution system. The average daily data from 14 February 2017 to 30 July 2017 are summarized. As can be seen from the waveform, the insulation resistance value is maintained at approximately 486 k$\Omega$ during the measurement period, indicating alarms and faults did not occur. It was confirmed that the insulation resistance value was monitored in real time.

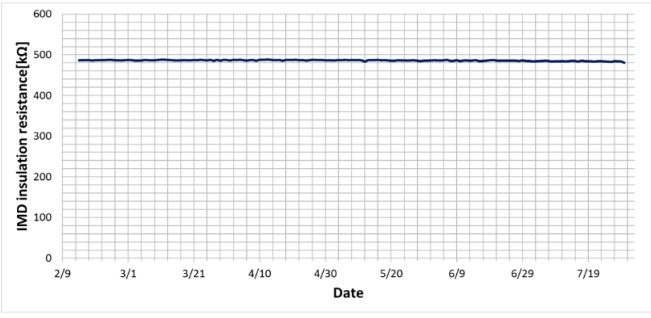

**Figure 11.** Measurement of insulation resistance.

The actual DC distribution system was completed in October of 2016 and continues to operate normally. The waveforms below in Figure 12 shows the operation data of the AC/DC Converter (input/output voltage, power) from 22 February 2017. As the load usage fluctuation is constant, a representative day's waveform was selected to be analyzed. The load usage ranged between 3 kW and 4 kW, with a sudden maximum peak at 9 kW. Although the load was not large, the protection operation of the power conversion device occurred intermittently in a sudden change of load condition that was not periodic.

The representative waveform at the time of the protection operation is shown in Figure 13. It is evident that the operation of the converter was interrupted at the moment when the protection operation was initiated. To understand this deeply, the current data at the time of the incident was acquired and analyzed with increased sampling time.

As shown in the waveform in Figure 13, 121 A of output current triggered the protection system. This exceeds the rated capacity of the converter and the threshold current of triggering protection system (Maximum 60 A) that indicates the need for more analysis.

A field survey confirmed that frequent water pump motor operations were found at farmhouses. The motor inrush current is commonly known to be 500%~700% of the rated current. However, in this incident, the inrush current reached 10 times the rated current. The DC/AC inverter system improvement plan is proposed as follows.

There was no inrush current problem in the operation of the DC distribution line since the protection operation level of the converter was reset. KEPCO Research Institute will continue to develop new problems and solutions through a distribution operation.

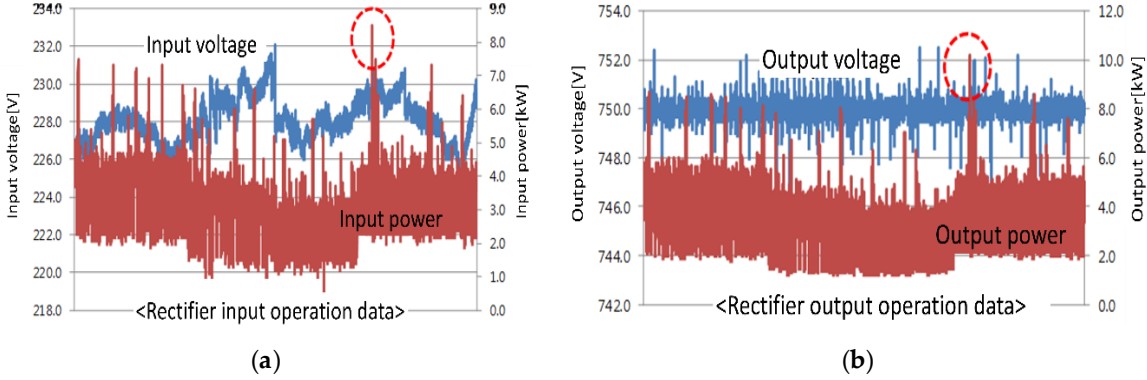

(**a**)　　　　　　　　　　　　　　　　　　　　　(**b**)

**Figure 12.** Input and output data of AC/DC Converter. (**a**) Input voltage and power. (**b**) Output voltage and power.

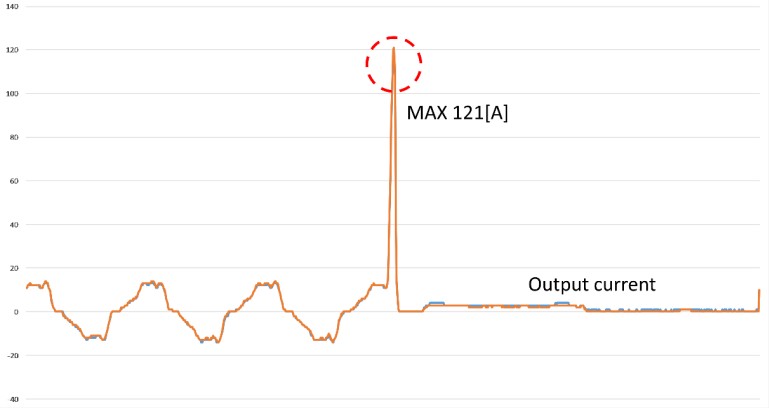

**Figure 13.** Waveform of protective action.

## 5. Conclusions

Based on the business model of the DC distribution system derived from the previous work, KEPCO research institute confirmed the possibility by applying DC distribution to the actual grid system. The power line subject to the test was set to Section 52L1–52L12 at Chungok Reservoir, a Gwangju Jeonnam HQ controlled area with the ABC cables. The target line is a long-distance low-load grid rural area with 12 spans or more. A 30 kW class AC/DC converter for supplying DC voltage and a 30 kW class DC/AC inverter for supplying AC voltage for household use were demonstrated to construct a DC distribution in the target line. In addition, an IT grounding system that is safe for electrolytic corrosion is applied to the DC distribution, and an IMD was installed to detect the failure situation in real time. Since the IMD is rarely used in distribution lines in general, KEPCO proposed the operational procedures and verified the feasibility of IMD by carrying out demonstration site tests accordingly. In addition to adopting the IT grounding system, the safety system of the DC distribution system was secured by designing and constructing the protection system.

By applying DC distribution to the actual grid system, the possibility of replacing the AC high voltage line to solve the operational problems such as contact with trees and to operate the stable line was confirmed. The established DC distribution is currently in operation. The operation data is and will be continuously analyzed to improve the deficiencies. In the future, KEPCO plans to construct an DC distribution system, with DC consumers, that will be more efficient and self-sustaining with connected distributed power resources.

**Author Contributions:** The following statements should be used "Conceptualization, J.K.; Methodology, J.K.; Software, H.K. (Hongjoo Kim); Validation, Youngpyo Cho, H.K. (Hyunmin Kim); Formal Analysis, J.C.; Investigation, Y.C.; Resources, H.K. (Hyunmin Kim); Data Curation, Y.C.; Writing-Original Draft Preparation, H.K. (Hyunmin Kim); Writin g-Review & Editing, J.K.; Visualization, J.C.; Supervision, J.K.; Project Administration, J.K.

**Funding:** This research was funded by Korea Electric Power Corporation (KEPCO) Research Institute grant number R15DA12.

**Conflicts of Interest:** The authors declare no conflicts of interest.

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
