# Peer review of "Application of a DC Distribution System in Korea: A Case Study of the LVDC Project"

_applsci, doi:10.3390/app9061074_

Round 1
Reviewer 1 Report
Compared to the literature, this paper applies DC distribution to real system operation. According to the authors, this has not been done before in the literature and all the experiments until now were performed in the lab.
The authors first design the DC distribution by specifying grounding, fault detection and protection. Then the perform tests and then a true experiment. This work has merit since DC distribution is of interest in some applications for energy efficiency. I think that the authors should address the following comments:
1) Explain better in the design part which are the differences with the AC distribution. What makes the DC distribution design challenging? For example, compared to AC distribution, AC circuit breakers are omitted and this renders protection challenges as new schemes should be developed. ABB has developed such a scheme. Could you compare with your approach? It would be useful to pinpoint similar challenges and perform such comparisons with AC distribution for the other parts of the design. Most readers are not familiar with DC distribution.
2) What are the challenging parts of inserting DC distribution in a real system? Is some similar effort done in the past? What could be the issues? It seems from the paper, that such a real-system implementation was easily done. Is this true and point where exactly is the value/merit of the paper?
3) In what scale does the DC distribution targets, e.g., microgrids or distribution grids?
3) The language needs improvement in several parts. Please check with an expert in English.
Author Response
1. In the design part, there are several differences with AC distribution. First we had to consider DC circuit breaker. It is more difficult to cut-off the DC line than AC line. So we have developed and installed DC circuit breaker which is suitable for LVDC. Next, grounding system has been considered. Non-grounding method(IT grounding) is used in DC distribution, but it is difficult to detect the failure situation of non-ground line due to its characteristics. Therefore, IMD devices for detection were added and verified through IMD operation methods and experiments.
2. It may seem easy to build, but research and development was not easy. As the application of DC distribution system was the first part of the challenge, there was a lot to consider. For example, IT grounding system and IMD operating precedure could be. The point of the paper is that DC distribution has been applied to the actual system.
3. Both microgrids and distribution grids can be targeted.
4. The linguistic parts of the paper were modified and supplemented.
Reviewer 2 Report
This paper presents the application of DC distribution system in long-distance low load rural area for utility grid. This document studies the problems caused by the operation of current DC distribution systems in rural environments. The abstract does not clearly present the description of the work developed by the authors. Neither does it indicate the contribution made by the authors in the section addressed. This summary does not include the main contributions and result obtained.
The introduction does not include the state-of-the-art of the problem studied. It would be advisable to increase the number of bibliographic references. As well as indicate the differences between the research works mentioned; showing some advantages and disadvantages between them. It may also be advisable to develop a comparison between the techniques applied in each case. It is recommended that the authors clearly indicate their contribution to the study with respect to other researches develops and cited in the references. In the introduction it is indicated that there is an improvement of the system, as well as a more effective model, but the criteria have not been developed in depth.
Section 2&3 shows the fundamentals of the problem addressed. The analysis of the models allows us to understand the criteria presented, although many of the concepts presented in this section are basic and could be part of the introduction of the paper. It is recommended to carry out a more in-depth analysis of section 3, in order that its analysis allows to understand the criteria and skills implemented. It would be advisable to increase the size of figure 8, in order to appreciate the details shown. The document presents some gaps in meaningful knowledge, as well as scientific character. There are no experimental results that support the results shown.
In some cases, the conclusions shown in the document are shallow. The included text is more related to a discussion of the topic addressed, than to the conclusions obtained according to the results presented. For this reason it is advisable to delve into these conclusions.
Most of the bibliographical references are current. All of them have been published in this last decade. Although it is advisable to increase their number. It would be advisable to carry out a search of more extensive academic work, all of them related to the topic addressed.
In short, in my opinion, the work here presented requires great changes. I hope that the comments provided serve to provide a new approach to the document. So that more detailed scientific explanations, along with the incorporation of experimental results that can support the approach shown.
Author Response
First of all, I am very grateful for your interest and feedback in my manuscript.
Also, I am very sorry for the late reply.
The amendment to the answer you sent is as follows.
1. I have increased the number of bibliographic references.
2. The size of figure 8 has been increased.
3. The state-of-the-art of the problem has been added in the introduction.
4. Abstract, Section 2&3 have been modified.
I hope that my review is adequate.

Reviewer 3 Report
This paper investigates the grounding systems, insulation and protection for a low voltage DC link based on a practical project. The paper can be a very valuable reference for other researchers due to the presentation of this real project and the field data. However, the writing should be further improved before being published. Some descriptions are not clear and misunderstanding. Moreover, there are typos and grammar mistakes. Please proofread the paper. The comments the reviewer wants to arise the attention of the authors are given below:
1. Page1, line 27, “power loss” should be “power losses”
2. Page 2, line 52, please provide full name for “IT, TN and TT”. Full name should be given for the first time use.
3. Page 4, line 149, the authors say the DCCB is installed in the system shown in Figure 5. Could the authors explain why it is needed? As the system is a point-to-point link, the ACCB can be used to isolate the DC side faults. The use of DCCB in a point-to-point system will significantly increases the capital cost as the cost of a DCCB is very high.
4. Please improve the quality of figures 3 and 4. They are not clear.
5. Page 6, 3. Field test, could the authors elaborate more about the system? Which types of ACDC converters (2-level VSC, 3-level VSC or MMC) and DCDC converters are used? Moreover, it will much grateful if the authors could provide a photo of the real system in your lab.
6. Page 6, (b) Case 1, could the authors clearly describe what has been carried out for Case 1? What is the fault applied in the system shown in Figure 7. Moreover, please describe what are these waves (there are 4 waves in each subfigure) shown in Figure 8? You cannot just put some results and let the authors guess what they are! Please give labels for each wave and put units in the Y-axis in the figures. Moreover, you need to clearly elaborate these waves and the reason behind them.
7. Page 7, line 197, “by applying a voltage to the line”, what is the voltage? Where it has been applied?
8. Page 7, line 201, which converter is in no-load condition? There are several converters in the figure! Why in no-load condition?
9. Page 8, 227, why faults caused by trees are reduced when the high-voltage line is replaced by low-voltage lines? the voltage level affects the trees? According to Figure 10, the AC high voltage line is higher than the low voltage DC line. Is it the reason the contacting tree faults are reduced?
10. Fig. 10 is not clear actually. Could the authors show system configurations using figures. Which type of converters is used in this practical project? 2L-VSC, 3L-VSC or MMC? What is the configuration of the converter station? Symmetrical monopole, asymmetrical monopole (with or without a metallic return?) or bipole ? What is the DC side grounding? DCCBs are employed in this system? How long is the link? These information is the most interesting for readers. The reviewer recommends the authors to use another figure like Figure 6 to clearly illustrate these information.
11. Page 8, line 240, according to the authors' descriptions, a new low-voltage line is built instead of utilizing the existing high-voltage ac line (conductors). It should not call “converts the existing 13.2 kV ac line to a dc” (line 222). Please double check.
12. Put units for the Y-axis in Figure 12.
13. Page 10, line 284, the overload triggered the protection system? please specify it.
14. Page 10, line 287, this description is not clear. What is the threshold of triggering the protection system? Please describe what actions have been taken when the protection is triggered. Figure 13 shows that the system was still operating when the overcurrent (121 A) appeared. According to the knowledge of the reviewer, the converter should be blocked immediately after detecting a fault, then the ac breaker will be tripped or the DCCBs will be tripped if they are used in this system. The descriptions about Figure 13 do not make sense for the reviewer.
15. The reviewer cannot get the point why there is a current of 242 A. What is this “output current”?
16. A suggestion for the authors that you may change the title of the paper to highlight this is a practical project. The review didt see the importance of “long-distance and low load” in this paper! You may use: Application of DC Distribution System in Korea: A Case Study of XXX Project. This may attract more views of your paper.
Author Response
First of all, I am very grateful for your interest and feedback in my manuscript.
Also, I am very sorry for the late reply.
The amendment to the answer you sent is as follows.
1. I have modified the word.
2. Full name for “IT, TN and TT” has been provided in a paper
3. The DCCB is installed in the system to protect converters.
4. The quality of figures 3 and 4 have been improved.
5. I have added table1 that includes what type of converter has been used. Also the photo of lab has been added.
6. Case 1 has been modified.
7. I have deleted the sentence "by applying a voltage to the line".
8. "AC/DC Rectifier(left side of the Figure 9)" is in no-load condition. I wanted to check the insulation resistance of IMD when the "AC/DC Rectifier" is in no-load condition. So we could compare with in load condition.
9. Lower voltages can reduce damage caused by trees than higher voltage. According to Korea's electricity standard law, electric poles cannot be raised as much as possible. It was also forced to be located below the AC high voltage line because the existing electric pole had to be used.
10. Fig.10 has been modified more bigger. 2-level VSC and symmetrical monopole have been used. As we mentioned in a paper, non-grounded systems have been used in DC. The line is about 600m.
11. I have changed that sentence.
12. I have put the units for the Y-axis in Figure 12.
13. The overload did not triggered the protection system. See also Fig 13 for information about how the protection system was triggered.
14-15. I've corrected the confusing part.
16. The title of the paper have been changed.
I hope that my review is adequate.

Round 2
Reviewer 2 Report
The authors have introduced most of the observations made by the reviewer. Although the abstracts continues without clearly indicating the contributions made by the authors to the research topic. The introduction also does not include the main contributions and results obtained. Some details of the figures continue to be blurred (for example figures 5, 8 and 13), please redo these diagrams.
The document presents some gaps in meaningful knowledge, as well as scientific character. There are no experimental results that support the results shown.
Author Response
Thank you for your sincere reply.
Also, I am very sorry for the late reply.
The modified parts are as follows.
1. I've modified some details of the figures.
2. Abstract and instruction parts have been modified.
I hope that my review is appropriate.
Thank you.

Reviewer 3 Report
Thanks for considering the comments from the reviewer. However, the responses are not good. Some of the questions are not clearly explained. The manuscript is not modified accordingly as well. For instance, the response 9 should be reflected in the manuscript not only by answering reviewer’s question. There were few questions in reviewer’s 14-15. However, they are not answered one by one. The cover letter should be the things you have changed point-by-point not just attaching the new manuscript. The work in the paper is interesting because it is a real-system. Otherwise, the paper will not be accepted with this level of writing and presentation. I also fully agree the comments from the other reviewer. The authors are encouraged to further modify the paper by seriously considering the comments from both reviewers 1 and 2 point-by-point.
Author Response
Thank you for your sincere reply.
Also, I am very sorry for the late reply.
The modified parts are as follows.
1. I have reflected in the manuscript not only by answering reviewer’s question.
2. To answer the reviewer's question 14, The threshold value was added to the paper. Also I have changed the figure 13 that confused readers.
3. To answer the reviewer's question 15, The sentence was modified by the author's misjudgment.
I hope that my review is appropriate.
Thank you.
